# Diagnosis of Sickle Cell Disease and HBB Haplotyping in the Era of Personalized Medicine: Role of Next Generation Sequencing

**DOI:** 10.3390/jpm11060454

**Published:** 2021-05-23

**Authors:** Adekunle Adekile, Nagihan Akbulut-Jeradi, Rasha Al Khaldi, Maria Jinky Fernandez, Jalaja Sukumaran

**Affiliations:** 1Department of Pediatrics, Faculty of Medicine, Kuwait University, P.O. Box 24923, Safat 13110, Kuwait; jalajasukumaran@hotmail; 2Advanced Technology Company, Hawali 32060, Kuwait; nagihanakbulut@atc.com.kw (N.A.-J.); rasha.m@atc.com.kw (R.A.); maria.f@atc.com.kw (M.J.F.)

**Keywords:** sickle cell diagnosis, haplotypes, next generation sequencing

## Abstract

Hemoglobin genotype and HBB haplotype are established genetic factors that modify the clinical phenotype in sickle cell disease (SCD). Current methods of establishing these two factors are cumbersome and/or prone to errors. The throughput capability of next generation sequencing (NGS) makes it ideal for simultaneous interrogation of the many genes of interest in SCD. This study was designed to confirm the diagnosis in patients with HbSS and Sβ-thalassemia, identify any ß-thal mutations and simultaneously determine the ß^S^ HBB haplotype. Illumina Ampliseq custom DNA panel was used to genotype the DNA samples. Haplotyping was based on the alleles on five haplotype-specific SNPs. The patients studied included 159 HbSS patients and 68 Sβ-thal patients, previously diagnosed using high performance liquid chromatography (HPLC). There was considerable discordance between HPLC and NGS results, giving a false +ve rate of 20.5% with a sensitivity of 79% for the identification of Sβthal. Arab/India haplotype was found in 81.5% of β^S^ chromosomes, while the two most common, of the 13 β-thal mutations detected, were IVS-1 del25 and IVS-II-1 (G>A). NGS is very versatile and can be deployed to simultaneously screen multiple gene loci for modifying polymorphisms, to afford personalized, evidence-based counselling and early intervention.

## 1. Introduction

Hemoglobinopathies are the most common monogenic disorders worldwide, with an estimated >300,000 affected infants born annually [1]. They are caused by mutations in the globin genes responsible for the production of the various chains of the hemoglobin (Hb) molecule. The latter is a tetramer of two pairs of α-like and β-like globin chains; the normal adult HbA is α2β2, the minor HbA2 is α2δ2, while the major Hb in fetal life is HbF, α2γ2. The structural hemoglobin variants result from point mutations, while synthetic abnormalities, referred to as thalassemias, usually result from point mutations and deletions (especially α-thalassemias). A few variants, e.g., HbE, combine both structural and synthetic defects [2].

The most important of the clinically-significant structural variants is HbS, which results from the HBB; glu6(E)val(A): GAG-GTG; rs334 mutation. Homozygosity (HbSS) is the most common genotype, resulting in sickle cell anemia (SCA), which is the most severe in the sickle cell disease (SCD) spectrum. Other forms are compound heterozygotes of HbS with other abnormal ß-chain variants, e.g., HbSC and HbSβ-thal. The clinical phenotype of SCD is characterized by recurrent vaso-occlusion and chronic hemolytic anemia. However, there is also a chronic inflammatory, pro-coagulant state, thus producing an extensive variety of down-stream, end-organ pathologies [3,4].

The diagnosis of SCD and other hemoglobinopathies has traditionally depended on hematological screening with complete blood count (CBC), RBC morphology, and Hb electrophoresis to separate Hb variants by their electrical charges and migration in an alkaline or acidic medium. Isoelectric focusing (IEF) is another useful method of electrophoresis. High performance liquid chromatography (HPLC) or capillary electrophoresis (CE) is used to determine the relative concentrations of different Hb fractions.

These methods are, however, fraught with ambiguities, especially in defining the carrier state or compound heterozygotes. Many Hb variants have normal CBC and RBC indices, while thalassemia carriers have microcytic, hypochromic anemia, which is also seen in iron deficiency. Moreover, Sβ-thal and HbSS patients with the α-thal trait both present with microcytosis and hypochromia. The distinguishing laboratory feature is the level of HbA2, which is usually >3.5% in the β-thal trait. Unfortunately, when HbA2 is estimated with HPLC in patients with Sβ-thal, it tends to be spuriously high because it co-migrates with glycated HbS ([5]). This becomes an issue in parts of the world, especially the Mediterranean and the Middle East, where both the HbS and β-thal traits are prevalent and compound heterozygotes form a significant proportion of the SCD population [6,7]. A combination of HPLC and DNA studies have shown that Sβ-thal constitutes about 40% of the patients with SCD in Kuwait [8,9].

Quite often, therefore, molecular studies are necessary to confirm a diagnosis ([10]). This is also mandatory for prenatal diagnosis, whereby parental mutations are ascertained, for definitive screening of a fetus, using several DNA methods, ranging from PCR to sequencing. What makes these molecular studies more fascinating is the discovery of genetic modifiers, which modulate the SCD phenotype and predispose it to, or protect it from, different complications. Thus, the phenotype in SCD is quite variable. Therefore, for complete and comprehensive counselling, it is necessary to determine each patient’s genotype, taking into consideration the different polymorphisms that significantly modify the phenotype. This is more so the case in the era of personalized medicine. The benefits of this detailed screening can only be maximized if offered to children as early as possible after their SCD diagnosis, so that an individualized management plan can be designed for each patient.

Two important recognized cis-acting modifiers of SCD phenotype are HbF and the β^S^ HBB gene cluster haplotype. HbF impedes the polymerization of HbS, leading to a significant amelioration in the clinical course of the disease [11,12]. While a normal individual has <1% HbF by the age of 6 months, the level remains persistently high in many patients with SCD [13]. The eventual concentration is under genetic control with the main cis-acting elements residing in the HBG2 promoter region, especially the –158 (C > T) XmnI polymorphism, rs7482144. GWAS studies have, however, revealed other significant trans-acting quantitative trait loci (QTLs) in the *BCL11A* gene and *HBS1L-MYB* intergenic region located on chromosomes 2 and 6, respectively [14,15].

There are five β^S^ HBB haplotypes, which are named for the parts of Africa and Asia where they predominate: Benin (BEN), Senegal (SEN), Arab/India (AI), Bantu (BAN) and Cameroon (CAM) [16]. They are of prognostic value with patients carrying the AI and SEN haplotypes having elevated HbF levels and a relatively mild phenotype [17,18]. BAN is associated with the most severe phenotype, while BEN and CAM have intermediate phenotypes [19]. These haplotypes are determined by the pattern of restriction fragment length polymorphisms (RFLPs) in the HBB gene cluster, following digestion with eight restriction enzymes [20]. An alternative method of haplotyping is fluorescence resonance energy transfer coupled with high resolution melting assay [21]. However, these two methods are labor intensive and also have other limitations. More recently, Shaikho et al. have utilized GWAS data and NGS to genotype several specific allele SNPs to derive the β^S^ HBB haplotypes among American and Saudi patients with SCD [22].

The traditional molecular techniques for the identification of HBB point mutations are PCR-based, followed by allele-specific oligonucleotide probe hybridization, amplification of the refractory mutation system, and restriction endonuclease analysis of PCR products. Gap-PCR protocols were developed to detect some of the more common globin gene cluster deletions [23,24]. Direct sequencing, using the Sanger method has been automated and widely used, while multiplex ligation probe amplification was developed to identify deletions and duplications [10]. Microarray-based comparative genome hybridization (aCGH) is another advancement in the techniques used for characterizing changes in copy numbers due to deletions and duplications for the identification of the so-called copy number variants (CNVs) [25].

The advantage of NGS over Sanger in the search for novel mutations is that, while the latter can read up to 500 nucleotides per reaction, NGS allows throughput sequencing of massive amounts of DNA [10,26,27]. It is possible to sequence a whole genome or targeted regions. This is very useful for investigating SCD, incorporating the possible loci for accurate diagnosis, and in the interrogation of phenotype-modifying loci. NGS has not been widely deployed in SCD research other than for extended blood group genotyping [28,29].

We have subjected archival DNA samples from our patients with HbSS and HbSβ-thal to NGS to confirm the Hb genotypes obtained with HPLC. We also simultaneously determined the β-thal mutations in the Sβ-thal compound heterozygotes and genotyped SNPs in the β-globin locus to establish HBB haplotypes. Other QTLs on chromosomes 2 and 6 were interrogated to investigate their association with HbF expression among our patients. However, this paper is limited to only the role of NGS in diagnosis and haplotyping.

## 2. Materials and Methods

The hemoglobin research laboratory in the Department of Pediatrics, Kuwait University, receives blood samples from both pediatric and adult hematology clinics in Kuwait. We have built up a repository of DNA samples from blood specimens, which have been subjected to various hematological and molecular investigations between 1994 and now. Many, but not all, had complete blood counts but all had cation-exchange HPLC (Shimadzu LC-20AT, Shimadzu Corporation, Kyoto, Japan) to quantify their Hb fractions, done in the referring hospitals or in our lab. The study was approved by the Human Research Ethics Committee of the Faculty of Medicine, Kuwait University and the Ethics Committee of the Ministry of Health, Kuwait.

The samples for the present study were selected from the repository and the only inclusion criteria were diagnosis (by the referring physician) of SCD, irrespective of Hb genotype, and availability of HPLC data. Genomic DNA was extracted from peripheral leukocytes using the phenol-chloroform method. The Illumina Ampliseq custom DNA panel was used to genotype the DNA samples. The panel contained SNPs from the following: HBB locus (HBE1, HBG2, HBG1, HBBP1, HBD and HBB), BCL11 and HBS1L-MYB intergenic region. All β-globin mutations and variants were confirmed by arrayed primer extension (APEX) [30] or Sanger sequencing methods. In addition to rs7482144, 58 other SNPs were genotyped, covering different regions of the HBB locus on chromosome 11.

The Hb genotype diagnosis by HPLC was compared to what was obtained by NGS to determine the rates of false positive results for Sβ-thal. The mean HbF and RBC indices were compared between SS and Sβ-thal patients.

β^S^ HBB haplotypes were determined using a modification of the SNP-based method described by Shaikho et al. [22]. We used the following 5 SNPs: rs968857, rs10128556, rs28440105, rs7482144, and rs3834466, as shown in Table 1. All the SNPs were tested for Hardy–Weinberg equilibrium and HaploView 4.2 software was used to classify the haplotypes.

## 3. Results

### 3.1. Diagnosis of Sβthal

DNA samples were from 240 patients, aged between 1 and 59, with a mean age of 12.7 ± 11.2 years. Table 2 shows the initial HPLC diagnosis and the eventual NGS results. Indeed, Sβ-thal constituted 22.5% in the former case, but in the latter, it was 28.3%. Three individuals who were diagnosed as HbSS turned out to be AS; one turned out to have HbAA and one had β-thal trait. Fourteen patients that were originally thought to have Sβ-thal turned out to have HbSS, i.e., a false positive rate of 20.5% and a sensitivity of only 79%. Table 3 shows the mean hematological values for the HbSS and Sβ-thal patients. There were significant differences (*p* value < 0.05) in MCV, MCH, and MCHC, but not in Hb and HbF.

### 3.2. Identification of β-Thal Mutations

A total of 13 mutations were identified in 66 Sβ-thal patients (Table 4). Of these, the most common were β^0^ IVS-1 del25 and IVS-II-1 (G→A) in 10 (15.2%) and 9 (13.6%) patients, respectively. The most common β+ mutations were IVS-I-110 (G→A) and IVS-I-5 (G→C), found in 8 (12.1%) patients each. While Sβ^0^-thal patients had no HbA on HPLC, Sβ^+^ mutations were associated with varying concentrations of HbA, ranging from 0 (for those carrying the IVS-I-5 (G→C) mutation) to a mean of 15.5 +/– 6.2% (in patients with IVS-I-110 (G→A)) and 25.1% in the Sβ^++^ patient with IVS-I-6 (C→T).

### 3.3. β^S^ HBB Haplotypes

Of the 154 HbSS patients that were successfully haplotyped, the most common case was homozygosity for AI, in 64.9% of patients, followed by compound heterozygosity for AI and an atypical haplotype in 16.2% of patients. The other rarer haplotypes are shown in Table 5. Among the Sβ-thal patients, the S chromosome was on an AI haplotype in 91.4% of cases (not shown in the table). The total number of S chromosomes in which HBB haplotypes were determined was 378, therefore giving a frequency of 81.5% for AI (TTCA2), 5.8% for BEN (TTCG1), 2.4% for SEN (TTCA1), 1.3% BAN (CCCG1), 2.1% CAM (TCAG1), and 6.9% for atypical haplotypes. The three prevalent patterns among the atypical haplotypes were CCCG2, TTCG2, and TCCG2, which we have termed ATP-I, ATP-II, and ATP-III, respectively. The mean age and hematological values among the different haplotype groups are shown in Table 5. Patients carrying the AI and SEN haplotypes had the highest mean HbF levels of ~24.0%. However, when the means were compared between the five groups carrying AI and the two carrying SEN, using ANOVA, there was no statistically significant difference. Atypical haplotypes were mostly co-inherited with the AI haplotype, as shown in the table. Indeed, the hematology and HbF levels were not significantly different in the latter case compared to AI homozygotes.

## 4. Discussion

The hallmark of SCD is its unyielding heterogeneity, which is mostly genetically driven, although environmental factors play some roles [31,32]. A myriad of genetic modifiers is being identified and it is increasingly possible, and desirable, to prognosticate the clinical phenotype, depending on the genotype of the patient at various cis- and trans-acting loci. Thus, the phenotype has a multi-genic basis and, in this era of personalized medicine, relevant panels for screening patients with SCD need to be developed to offer more evidence-based counseling at an early age. This is even more relevant in populations with significant genetic admixtures as is prevalent in the Arabian Peninsula. The growing availability of NGS provides a platform that serves this purpose, making it possible to interrogate thousands of genes in a parallel run. It has a tremendous potential for use among infants and young children with SCD.

The diagnosis of sickle compound heterozygotes is notoriously difficult and, quite often, has to be resolved with molecular techniques. In Kuwait and Saudi Arabia, unlike in Africa and the Western World, over a third of patients with SCD have the Sβthal genotype [6,9]. Hb electrophoresis and HPLC are not adequate to accurately identify these patients and this is well illustrated in the present study, with a substantial discordance between HPLC and NGS diagnoses. In situations where molecular studies are not available for resolution, the need for family studies cannot be over-emphasized. This is the practice in pediatric SCD clinics, such that if one parent does not carry the sickle gene, then it is unlikely that the index patient has HbSS. This is not always possible when patients are seen for the first time as adolescents or adults. RBC indices may also be helpful, but where a co-existent α-thalassemia trait and iron deficiency are common, they constitute cofounding factors.

The pattern of β-thal mutations in this study is similar to that in our previous report ([8]), but this is the first report of CD 82/83 (-G) deletion among Kuwaitis. Although it has been previously reported in Azerbaijan and Czechoslovakia [33,34], we are not aware of any reports from the Arabian Gulf region. While the IVS-I-5 (G→C) mutation is classified as a β+ trait, it is so severe that, frequently, as in this report, there is no identifiable HbA on HPLC in patients with Sβ-thal carrying the mutation. That is why we combined it with the Sß0 group in our previous study [8]. A consensus may need to be reached on how to classify these patients in the future. The fact that most of our Sβ+ patients carry either IVS-I-110 (G→A) or IVS-I-5 (G→C) makes the condition quite severe and often, indistinguishable from the Sβ0 or SS phenotypes. On the other hand, patients carrying the β^++^, IVS-I-6 (C→T) mutation have a milder phenotype, comparable to HbAS [8]. For this reason, it is advisable to confidently establish a molecular diagnosis for individualized care and counseling for patients with Sβ-thal.

Haplotyping of β^S^ HBB has clinical, hematological, and anthropological implications, but previous methods of determination were rather complicated. NGS has considerably simplified the procedure in a reliable and reproducible way; thus, we were able to refine the haplotype pattern in our patients. The majority of the patients are homozygous for the AI haplotype, but all other haplotypes, including atypicals, are present, albeit at low frequencies. While the mean HbF levels were highest in AI homozygotes, all the other haplotypes, including the BEN and CAM homozygotes, were also associated with relatively higher (>10%) levels than in African patients with similar haplotype backgrounds. This corroborates the findings in other studies in the Arabian Peninsula [35,36,37], suggesting the presence of compensatory SNPs that drive HbF expression in this population. The pattern of the prevalent atypical haplotypes among our patients is interesting, with the ATP-I haplotype being more common than some of the typical patterns. It will be enlightening to determine the frequencies of these haplotypes among other Arab Gulf states. In a separate study, we shall report the relative frequencies of atypical haplotypes and their contributions, if any, to clinical and hematological phenotypes.

One important utilization of NGS in SCD is the search for novel SNPs that modify the phenotype in the disease, especially those driving HbF expression. Apart from cis-acting SNPs on chromosome 11, the other QTLs on chromosomes 2 (BCL11A) and 6 (*HBS1L-MYB*) have shown strong associations with HbF [14,15]. Past studies have demonstrated that many of the SNPs that are important in other populations, in this regard, are not present among Gulf Arab patients [38,39,40]. Our data on SNPs in these QTLs are being analyzed and will be reported shortly.

Apart from those associated with HbF expression, many other modifiers have emerged from genotype–phenotype association studies, and, lately, from GWAS. Some of these have been validated, but many more still await larger studies. The alpha thalassemia trait is associated with an amelioration of many of the complications of SCD, including stroke, priapism, splenic dysfunction, and leg ulcers [41,42,43,44]. SNPs in the ANXA2, TEK and TGFBR3 gene loci were associated with stroke risk [32,45,46,47], the TGF-β/BMP pathway is associated with several SCD subphenotypes [48,49,50], while polymorphisms within the UGT1A1 are associated with increased serum bilirubin levels and cholelithiasis [51]. In the near future, it will be possible to simultaneously screen for many of these SNPs, in addition to confirming a diagnosis of SCD. NGS is a promising platform in this regard.

## Figures and Tables

**Table 1 jpm-11-00454-t001:** Alleles of the 5 SNPs used to define the common haplotypes.

Haplotype	rs968857	rs10128556	rs28440105	rs7482144	rs3834466 *
AI	T	T	C	A	2
SEN	T	T	C	A	1
BEN	T	C	C	G	1
BAN	C	C	C	G	1
CAM	T	C	A	G	1
** ATP-I	C	C	C	G	2
** ATP-II	T	T	C	G	2
** ATP-III	T	C	C	G	2

* 1: T, 2: dupT [TT], ** Atypical haplotypes.

**Table 2 jpm-11-00454-t002:** Distribution of Hb genotypes of patients in the study.

Group	HPLC	NGS
SS	184	159
Sβ	54	68
SD	2	4
AS	0	7
AA	0	1
β trait	0	1
Total	240	240

**Table 3 jpm-11-00454-t003:** Mean age and hematological values in HbSS- and HbSβ-thalassemia patients.

Group	Age (years)	* Hb g/dl)	** RBC (×10^9^/l)	*** MCV (fl)	**** MCH (pg)	$ MCHC (g/dl)	$$ HbF (%)
SS	13.3+/−11.7	11.2+/−9.7	3.8+/−0.9	80.6+/−13.4	27.2+/−4.5	33.4+/−1.4	22.6+/−8.5
Sβ	11.2+/−9.7	10.7+/−0	4.4+/−0.8	69.5+/−8.8	22.4+/−2.8	32.2+/−1.6	24.7+/−11.2

* Hemoglobin. ** Red blood cell. *** Mean corpuscular volume. **** Mean corpuscular hemoglobin. $ Mean corpuscular hemoglobin concentration. $$ Fetal hemoglobin.

**Table 4 jpm-11-00454-t004:** Distribution of beta-thalassemia mutations in Sβ patients and the associated mean hematological values.

Legacy Name (Nucleotide Location)	HGVS Nomenclature (Nucleotide Change)	β^0^/β^+^	N (%)	Hb g/dl	RBC X 10^9^/l	MCV fl	MCH pg	MCHC pg/l	HbF %
IVS-1, -25del	c.93-22_95delbp	β^0^	10 (15.2)	9.4 +/−1.7	4.6+/−0.7	65.8+/−3.6	20.5+/−0.9	31.1+/−0.7	29.6+/−11.7
IVS-II-1	c.315+1G>T	β^0^	9 (13.6)	10.9+/−1.5	4.8+/−0.7	66.9+/−2.4	22.6+/−0.8	32.7+/−1.6	33.5+/−6.0
IVS-I-110	c.93-21 G>A	β^+^	8 (12.1)	9.4+/−1.1	4.2+/−0.7	69.9+/−5.6	22.5+/−1.7	32.3+/−1.2	12.6+/−4.9
IVS-I-1	c.92+1G>T	β^0^	8 (12.1)	9.6+/−1.6	4.3+/−0.6	70.3+/−8.4	22.3+/−1.7	31.6+/−1.6	28.3+/−9.9
IVS-I-5	c.92+5 G>C	β^+^	8 (12.5)	9.2+/−0.7	4.5+/−0.8	67.4+/−4.9	21.4+/−2.0	31.8+/−1.6	20.6+/−13.9
CD39	c.118 C>T	β^0^	5 (7.2)	9.1+/−0.9	3.6+/−0.4	82.7+/−16.0	25.7+/−5.0	31.5+/−0.5	20.1+/−12.9
CD8	c.25_26delTT	β^0^	5 (7.2)	9.2+/−1.1	4.5+/−0.7	61.3+/−7.0	20.8+/−0.4	34.1+/−1.8	32.3+/−7.7
CD8/9	c.27dupG	β^0^	3 (4.5)	8.1+/−0.9	3.7+/−0.7	72.5+/−2.7	23.0+/−0.9	31.6+/−0.1	18.9+/−1.7
CD36/37	c.112 delA	β^0^	3 (4.5)	8.4+/−0.6	3.2+/−0.6	63.4+/−2.3	24.3+/−3.0	32.8+/−1.4	14.2+/−4.5
CD31	c.92G>C	β^0^	3 (4.5)	10.1+/−1.8	4.9+/−1.0	66.5+/−2.1	20.8+/−0.4	31.3+/−0.5	25.6+/−2.7
CD41/42	c.126_129delCTTT	β^0^	2 (3.0)	11.6+/−0.2	4.8+/−0.5	76.8+/−11.0	24.4+/−2.2	31.8+/−1.7	33.9+/−13.6
IVS-I-6	c.92+6 T>C	β^+^	1 (1.5)	11.1	5.1	66.7	21.6	32.4	8.5
CD 82/83	c.251delG	β^0^	1 (1.5)	9.0	4.2	64.0	21.3	33.3	15.4

**Table 5 jpm-11-00454-t005:** Haplotypes among HbSS patients.

	Haplotype	n (%)	Age (Years)	* Hb (g/dl)	** MCV (fl)	*** MCH (pg)	**** HbF (%)
**SS**	AI/AI	100 (64.9)	13.2+/−11.8	11.2+/−7.5	81.0+/−13.8	27.5+/−4.3	24.2+/−7.9
	AI/ATP	25 (16.2)	15.0+/−13.2	9.9+/−9.2	76.6+/−12.1	25.9+/−4.6	21.0+/−7.9
	AI/BEN	9 (5.8)	15.3+/−13.7	9.4+/−1.5	86.1+/−15.6	29.0+/−6.8	19.3+/−11.5
	AI/SEN	5 (3.2)	12.1+/−8.3	10.7+/−0.5	65.2+/−4.1	21.1+/−1.4	15.2+/−8.8
	BEN/BEN	4 (2.6)	6.4+/−5.0	8.9+/−1.0	81.9+/−8.5	26.9+/−4.0	16.8+/−9.2
	AI/CAR	4 (2.6)	12.0+/−12.2	10.0+/−0.4	90.2+/−8.6	29.8+/−3.0	19.7+/−5.1
	CAM/CAM	2 (1.3)	14.5+/−4.9	8.9+/−0.1	89.5+/−19.2	29.2+/−7.1	13.2+/−2.0
	SEN/SEN	2 (1.3)	16.0+/−7.1	10.7+/−1.1	85.2+/−21.0	29.2+/−8.5	24.1+/−11.0
	BEN/CAM	1 (0.6)	11	11	78.6	26	34.9
	BEN/ATP	1 (0.6)	7	7.9	76	24	9.3
	AI/CAM	1 (0.6)	2	9.7	68	31.3	22.6

* Hemoglobin. ** Mean corpuscular volume. *** Mean corpuscular hemoglobin. **** Fetal hemoglobin.

## Data Availability

All data are available on demand from the first author.

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
