# Peer review of "Diagnosis of Sickle Cell Disease and HBB Haplotyping in the Era of Personalized Medicine: Role of Next Generation Sequencing"

_jpm, 2021, doi:10.3390/jpm11060454_

Round 1

Reviewer 1 Report

In this study, the authors aim to confirm by NGS the diagnosis previously performed by HPLC in patients with HbSS and Sβ-thalassemia, and simultaneously to determine the ßS HBB haplotypes using five specific SNPs.

This a valuable and interesting well-written manuscript. However, in my opinion, the study could be improved with a more powerful statistical analysis, testing the association between individual SNPs and derived haplotype with HbF levels in the two main groups of patients (SS and S β patients).

Minor points:

Tables 3 and 4: The abbreviations should be included regarding the hematological parameters. The same in Table 5, including for the haplotype nomenclature.

Subsection 3.3, Lines 219-221, the authors state that “Patients carrying the AI and SEN haplotypes had the highest mean HbF levels of ~24.0%. However, when the means were compared by ANOVA, there was no statistically significant difference.” Clarify the performed comparison with ANOVA. It was between AI and SEN haplotypes?

Author Response

We thank the reviewer for the kind words and comments. Our responses are given below

Comment: This is a valuable and interesting well-written manuscript. However, in my opinion, the study could be improved with a more powerful statistical analysis, testing the association between individual SNPs and derived haplotype with HbF levels in the two main groups of patients (SS and Sβ patients).

  • Response: We thank the reviewer for the kind words. We did not think further statistical analysis was warranted especially because of the small numbers of patients in each of the haplotype groups. However, further testing was done in our separate report on > 130 SNPs in the modifier genetic loci and their association with HbF levels

Minor points:

Comment: Tables 3 and 4: The abbreviations should be included regarding the hematological parameters. The same in Table 5, including for the haplotype nomenclature.

  • Response: Abbreviations of the hematological parameters have been provided as footnotes

Comment: Subsection 3.3, Lines 219-221, the authors state that “Patients carrying the AI and SEN haplotypes had the highest mean HbF levels of ~24.0%. However, when the means were compared by ANOVA, there was no statistically significant difference.” Clarify the performed comparison with ANOVA. It was between AI and SEN haplotypes?

Response: This has been clarified to indicate that ANOVA was done to compare mean HbF levels between the 5 groups carrying AI and the 2 carrying SEN haplotypes

Reviewer 2 Report

In our era the NGS is a very important tool for identification of genomic variants, SNPs e modifier genes for the severity of the Sickle Cell Disease. This allows us to get to personalized medicine.

The authors have clearly analyzed every aspect proposed in the study design.

I have some minor comments:

  • Line 112: the latter can only read up to 100 nucleotides per reaction”…This sentence is not correct: the Sanger can read up to 500 nt. Please rewrite the sentence and specify.

  • Line 136 : the choice of genes needs to be better discussed. Please specify the number of the genes in the custom panel. Is present the KLF1 gene?

  • Line 142: Please discuss also the value of HbA2 for identification beta-thal patients for the

  • Line 298: To cite the paper: Maddalena Martella, Nadia Quaglia, Anna Chiara Frigo, Giuseppe Basso, Raffaella Colombatti, Laura Sainati. “Association between a combination of single nucleotide polymorphisms and large vessel cerebral vasculopathy in African children with sickle cell disease”. Blood Cells, Molecules, and Diseases. Vol 61, October 2016, Pages 1-3.

  • References: Please cite more recent references about NGS.

  • English: Please use words more appropriate for scientific field. For example:
  • line 261: “lumped”
  • line 266: “akin”
  • line 267: “firmly”
  • line 270: “cumbersome”

Author Response

We thank the reviewer for the kind words and comments. Our responses are given below

  • Comment: Line 112, “the latter can only read up to 100 nucleotides per reaction”…This sentence is not correct: the Sanger can read up to 500 nt. Please rewrite the sentence and specify.
    • Response: This has been corrected to reflect that Sanger can read up to 500 nt
  • Comment: Line 136, the choice of genes needs to be better discussed. Please specify the number of the genes in the custom panel. Is present the KLF1 gene?
    • Response: The genetic loci interrogated have been enumerated. KL1 was not included
  • Comment: Line 142, Please discuss also the value of HbA2 for identification beta-thal patients for the
    • Response: This was addressed in the introduction, lines 58 - 61
  • Comment: Line 298, To cite the paper: Maddalena Martella, Nadia Quaglia, Anna Chiara Frigo, Giuseppe Basso, Raffaella Colombatti, Laura Sainati. “Association between a combination of single nucleotide polymorphisms and large vessel cerebral vasculopathy in African children with sickle cell disease”. Blood Cells, Molecules, and Diseases. Vol 61, October 2016, Pages 1-3.
    • Response: This has been included as reference #25
  • Comment: References: Please cite more recent references about NGS.
    • We could not find many references on the use of NGS in sickle cell research except for a few on extended blood group genotyping. Two of these have been included as references #13 and #16
  • Comment: English: Please use words more appropriate for scientific field. For example:
  • line 261: “lumped”
    • Response: Changed to “combined
  • line 266: “akin”
    • Response: Changed to “comparable
  • line 267: “firmly”
    • Response: Changed to “confidently
  • line 270: “cumbersome”
    • Response: Changed to “rather complicated